# Lamina-specific neuronal properties promote robust, stable signal propagation in feedforward networks

**Dongqi Han**
Cognitive Neurorobotics Research Unit
Okinawa Institute of Science and Technology
Okinawa, Japan
dongqi.han@oist.jp

**Erik De Schutter**
Computational Neuroscience Unit
Okinawa Institute of Science and Technology
Okinawa, Japan
erik@oist.jp

**Sungho Hong**[*]
Computational Neuroscience Unit
Okinawa Institute of Science and Technology
Okinawa, Japan
shhong@oist.jp

## Abstract

Feedforward networks (FFN) are ubiquitous structures in neural systems and have been studied to understand mechanisms of reliable signal and information transmission. In many FFNs, neurons in one layer have intrinsic properties that are distinct from those in their pre-/postsynaptic layers, but how this affects network-level information processing remains unexplored. Here we show that layer-to-layer heterogeneity arising from lamina-specific cellular properties facilitates signal and information transmission in FFNs. Specifically, we found that signal transformations, made by each layer of neurons on an input-driven spike signal, demodulate signal distortions introduced by preceding layers. This mechanism boosts information transfer carried by a propagating spike signal and thereby supports reliable spike signal and information transmission in a deep FFN. Our study suggests that distinct cell types in neural circuits, performing different computational functions, facilitate information processing on the whole.

## 1 Introduction

How different cell types in a neural system contribute to signal processing by the entire circuit is a prime question in neuroscience. Experimental investigations of this question are increasingly common, primarily due to advances in observing and manipulating neurons based on their genetic signature. Feedforward circuits are notable targets of those studies, since, in many systems, they have been observed to comprise cell groups or "layers" with properties distinct from those of other layers, in size, morphology, expression of membrane/intracellular mechanisms, etc. For example, in the *Drosophila* antennal lobe (AL), projection neurons (PN) tend to show noisy firing, slow responses to the onset of olfactory receptor neuron (ORN) firing, and static voltage thresholds for spike generation whereas postsynaptic neurons of PNs in lateral horns (LHN) are less noisy, fire early, and have dynamical firing thresholds (Jeanne and Wilson, 2015). In the cerebellum, the granule cells are tiny neurons with simple morphology, but their postsynaptic targets, Purkinje cells, are large, with complex dendritic trees. Pre- and postsynaptic neurons having distinct intrinsic properties can be ubiquitously found in a wide variety of neural systems. These observations raise questions about the

---

[*]Corresponding author

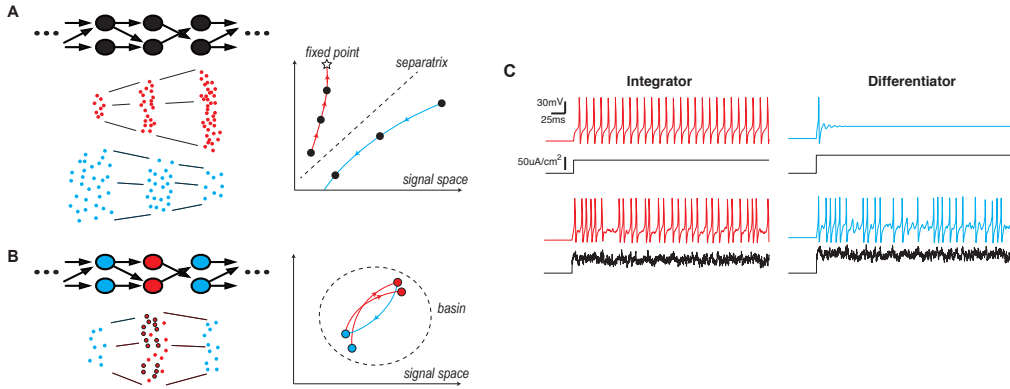

Figure 1: **Lamina-specific intrinsic properties stabilize information transmission in a neural network.** (A) Left: FFN with a single cell type (Top), and spikes at each layer, in two different modes of signal propagation (Bottom). One mode is amplification by progressively evoking more and more spikes (red dots), and the other is dissipation by gradually losing spikes (blue dots). Right: Trajectories of the two propagating signals in a signal space. The x- and y-axes represent independent signal characteristics, such as the number of spikes, temporal precision, etc. A star is a fixed point of neuronal signal transformation, and a dotted line is a separatrix separating the two modes. (B) Left: An FFN where neurons have lamina-specific intrinsic properties (Top). Each layer performs a layer-specific transformation, and can selectively transfer a subset of input spikes (circled red dots), ignoring those that cause signal distortion (Bottom). Right: Trajectory of a propagating signal in a signal space. The dotted circle surrounds a region (basin) where layer-specific transformations confine the propagating signal. (C) Behaviors of the two types of neurons used in this study with different dynamics of their spiking thresholds, shown by membrane potential response (color) to constant or fluctuating current injection (black).

role of intrinsic properties and their laminar specificity. However, most theoretical and computational studies rarely take neuronal heterogeneity into account.

We addressed this question by studying the classical problem of how a spike signal, defined by the evoked firing of multiple neurons in one layer, can stably propagate through multiple downstream layers in an FFN (Diesmann et al., 1999; van Rossum et al., 2002; Reyes, 2003; Kumar et al., 2008a, 2010; Kremkow et al., 2010; Moldakarimov et al., 2015; Joglekar et al., 2018). FFN is an important model for reliable information transfer between distant brain regions, since, if a need for new functional connections arises due to learning, long-distance axons for direct connections cannot grow in adult brains but can be added only through evolutionary processes. Instead, the connectome data suggest that brain regions tend to organize into clusters to allow for strong *indirect* connectivity (Oh et al., 2014), which can form FFNs. Stable propagation of spike signals in FFNs also plays a key role in models of conscious perception (Joglekar et al., 2018; Van Vugt et al., 2018), learning in deep artificial networks (Samuel S Schoenholz and Sohl-Dickstein, 2017), etc.

Most of those studies assumed that FFNs have identical types of neurons, and thus each layer makes similar input/output transformations. In this case, an input-driven spike signal either gets stronger or weaker as it passes through layers, depending on the efficacy of output spike generation, given input spikes and also given the characteristics of the network input (Fig. 1(A), Left). Then, the signal eventually reaches a fixed point of layer-to-layer transformation or dissipates (Diesmann et al., 1999; Reyes, 2003; Moldakarimov et al., 2015) (Fig. 1(A), Right). In this scenario, stable signal transmission is achieved by specific conditions for a non-trivial fixed point, which are often not robust for a wide range of initial signals. Also, irreversible signal distortion during propagation can cause the inevitable dissipation of information.

Introducing lamina-specific intrinsic properties in neurons can change this fundamentally (Fig. 1(B)). If each layer transforms a propagating signal in a different direction than the previous one, a fixed point will not exist in general. Instead, this prevents the repeated transformation of the signal in one direction, and the overall signal distortion over multiple layers can become smaller, compared to

networks with identical layers. In particular, if the transformation by one layer is in the opposite or nearly opposite direction to those by its presynaptic layer, it can limit signal distortion across multiple layers and facilitate stable propagation (Fig. 1(B) Right). At the same time, information transfer also improves. Signal distortion at each layer will accumulate if neurons in each layer repeatedly encode similar preferred features of a network input, and this will cause irreversible loss of information. Contrarily, if the pre- and postsynaptic neurons encode distinct input features, more selective filtering on presynaptic output by postsynaptic neurons will demodulate distortions that presynaptic signal transformation introduced (Fig. 1(B) Left). In this manner, FFNs with heterogeneous, lamina-specific neuronal properties can show enhanced information transmission compared to homogeneous FFNs.

Here we demonstrate how robust, stable signal and information transmission arise from laminar specificity of cell-intrinsic properties by computational FFN models. We first introduce two types of neurons with different spiking dynamics, referred to as integrator and differentiator (Kumar et al., 2010; Ratté et al., 2013, 2015) (Fig. 1(C)). Using neural layers with such laminar specificity, we develop a model of the *Drosophila* AL network with three layers of ORNs, PNs, and LHNs. We show that this model replicates key findings of a recent experimental study (Jeanne and Wilson, 2015) that differences in spiking dynamics between PNs and LHNs can balance accuracy and speed in processing olfactory information, and furthermore that PN-to-LHN information transfer is nearly optimal. Then, we extend the model to a deep FFN and demonstrate robust and stable spike signal propagation, contrary to models with no laminar specificity in neuronal properties. Finally, we demonstrate that the speed of a propagating signal depends on the input signal property and, therefore, that deeper layers can use the latency coding for the input.

## 2 Related work

Many computational studies have explored how a spike signal can reliably propagate in spiking neural networks. Most of them hypothesize that the FFNs consist of homogeneous excitatory neurons. Earlier work (Diesmann et al., 1999; Kumar et al., 2008a, 2010; van Rossum et al., 2002; Vogels and Abbott, 2005) investigated conditions of stable signal transmission in multi-layer networks with purely feedforward connections, mostly as a form of synchronized spikes. These conditions involve input signal properties (Diesmann et al., 1999; Kumar et al., 2008a, 2010), noise in neural dynamics (van Rossum et al., 2002; Vogels and Abbott, 2005), and connection strength and sparsity (Kumar et al., 2008a, 2010). Recent studies also suggested that stable transmission can be enhanced by feedback connections (Moldakarimov et al., 2015; Joglekar et al., 2018).

While heterogeneity in synaptic properties are often studied, heterogeneity of cellular properties in neural networks has been less emphasized. Kumar et al. (2008a) model neurons in a group with random distributions of intrinsic properties, and many network models (Yarden and Nelken, 2017; Joglekar et al., 2018) introduces heterogeneity of different areas so as to match physiological findings. However, these studies have not discussed how heterogeneous neuronal properties affect the entire network. Our work is original as we focus on the role of laminar-specific neuronal properties in FFNs, which have been rarely investigated in previous studies, despite their ubiquity in neural systems. Our paper is the first one, to our knowledge, to computationally investigate the role of laminar-specific neuronal properties for reliable signal transmission through FFNs.

## 3 Methods

We used conductance-based model neurons based on the Morris-Lecar mechanisms (Morris and Lecar, 1981; Prescott et al., 2008), given by

$$C\frac{dV}{dt} = -g_L(V - E_L) - g_K w(V - E_K) - g_{Na} m_\infty(V)(V - E_{Na}) + I_{stoch} + I_{input},$$

$$\frac{dw}{dt} = \phi_w \frac{w_\infty(V) - w}{\tau_w(V)}, \quad z_\infty(V) = \frac{1}{2}\left[1 + \tanh\left(\frac{V - \beta_z}{\gamma_z}\right)\right] \quad (z = m, w),$$

$$\tau_w(V) = \cosh\left(\frac{V - \beta_w}{\gamma_w}\right)^{-1}, \tag{1}$$

where $V$ and $w$ are membrane potential and a gating variable for the $K^+$ channel. The first model, which we called the "integrator" neuron, had a high *half-maximum voltage* of the $K^+$ channel,

$\beta_w = 5$ mV. The other, "differentiator" neuron, had low $\beta_w = -19$ mV (See Section 4.1). A special case is the ORN in the *Drosophila* AL network model, which had $\beta_w = -23$ mV for stronger differentiator traits (Nagel and Wilson, 2011). The other parameters, which are the same as in Prescott et al. (2008), are shown in Appendix Table A1. Stochastic current $I_{stoch}$ represented noisy membrane potential fluctuation due to components that are absent in our model, such as background network inputs, an unknown noise source (Jeanne and Wilson, 2015), etc., and was given by an Ornstein-Uhlenbeck (OU) process, $dI_{stoch}/dt = -I_{stoch}/\tau_V + \sigma_V \xi$, where $\xi$ is a unit Gaussian noise, renewed each time step. $\tau_V = 1$ ms, and $\sigma_V$ was tuned to experimental data of Jeanne and Wilson (2015).

The input current $I_{input}$ was either synaptic inputs or a common current injection to input layer neurons. Each synaptic input were conductance-based and modeled as a double exponential function: at each presynaptic spike at $t_s$, the synaptic current was

$$I_{input}(t) = g(t)(V - E_{syn}), \quad g(t) = g_{syn}\left[e^{-(t-t_s)/\tau_1} - e^{-(t-t_s)/\tau_2}\right]H(t - t_s), \quad (2)$$

where $H(t)$ is a Heaviside function that $H(t) = 1$ if $t > 0$ and $H(t) = 0$ otherwise. We used $\tau_1 = 0.5$ ms and $\tau_2 = 4$ ms, tuned $g_{syn}$ to experimental data as $\sigma_V$. All other parameters are in Appendix Table A1. In the simulation of the optogenetic experiments in (Jeanne and Wilson, 2015), the input layer neurons were injected

$$I_{input} = I_{amp}[e^{(-(t-t_0)/\tau_{act})} - e^{(-(t-t_0)/\tau_{deact})}]H(t - t_0), \quad (3)$$

where $I_{amp}$=45 $\mu$A/cm$^2$, $\tau_{act}$=15 ms, and $\tau_{deact}$=50 ms. $t_0$=200 ms is a stimulus onset. See Appendix Table A2 for other parameters. We also performed a similar current injection to the input layer to generate dynamical spike signals (Fig. 2(D-E) and 3(D)). Here we used the OU process,

$$dI_{input}/dt = (\mu_{input} - I_{input})/\tau_{input} + \sigma_e \xi, \quad (4)$$

where $\mu_{input}$=15 $\mu$A/cm$^2$, $\sigma_{input}$=7.5 $\mu$A/cm$^2$, and $\tau_{input}$=5 ms.

All the simulations were constructed and run in Python using the Brian2 simulator (Goodman and Brette, 2008).

## 4 Results

### 4.1 Voltage-sensitive K$^+$ channels control dynamical input/output properties of neurons

A recent experimental study showed that PNs and their feedforward targets, LHNs, have different intrinsic properties in *Drosophila* AL network (Jeanne and Wilson, 2015). We created the computational models of those neurons, described by Eq. (1). Here a crucial parameter is $\beta_w$, the half-activation voltage of the K$^+$ channel. With lower $\beta_w$, the channel is more active at subthreshold voltages, shifting the balance between inward and outward currents around the spiking threshold. Then, the computational property of the neuron profoundly changes with strengthened differentiator-like traits, whereas higher $\beta_w$ promotes integrator-like behavior (Prescott et al., 2008; Ratté et al., 2013). For example, typical repetitive firing with a sustained current input, seen in the integrator neurons with high $\beta_w$, was suppressed in those with the differentiators with low $\beta_w$. However, the differentiators showed robust sensitivity to the dynamic fluctuation in inputs, demonstrated by evoked firing (Fig. 1(C)). Crucially, our integrator and differentiator model neurons showed experimentally observed differences in the spiking thresholds. The spiking threshold was significantly more dynamic in the differentiators, just as in LHNs, whereas the integrators tended to have the static voltage threshold, as in PNs (see Appendix A1). These suggested that the integrator and differentiator neurons can be good models for *Drosophila* PNs and LHNs, respectively.

### 4.2 Lamina-specific neuronal properties are crucial for the *Drosophila* AL network

Using the model neurons, we developed a network model of the *Drosophila* AL (see Appendix A1 for parameters). 40 ORNs projected to each PN, and nine PNs projected to each of nine LHNs (Jeanne and Wilson, 2015). Each layer contained 100 replicas of these, corresponding to 100 "trials" of an experiment, resulting in 4,000 ORNs, 900 PNs, and 900 LHNs in an entire network. We tuned the parameters, such as injected current, synaptic conductances and $\sigma_V$ for each layer, to match

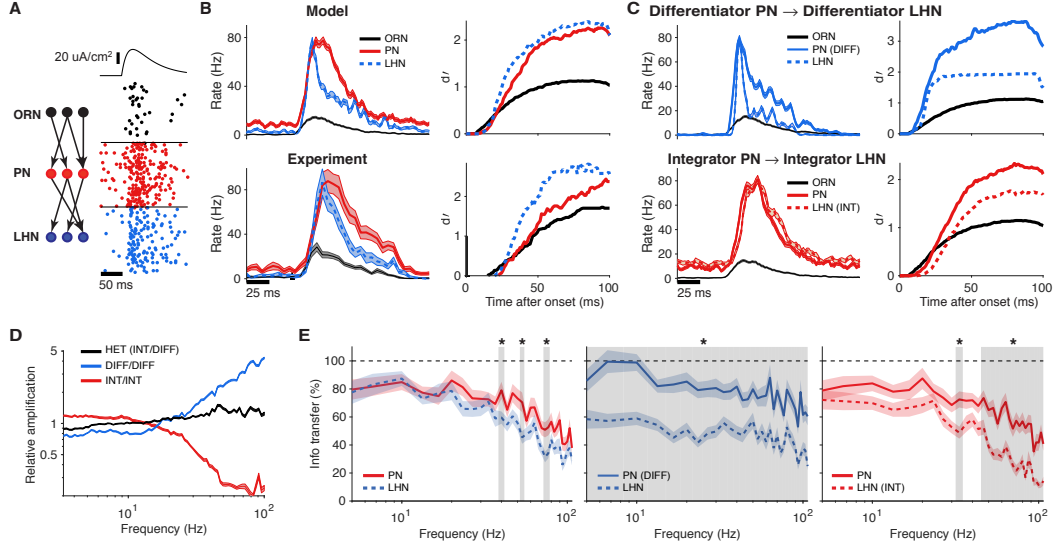

Figure 2: **Lamina-specific neuronal properties boost information transfer in the AL network.**
(A) Schematic diagram of the network model (Left), and spikes from a simulation with a current
input to ORNs on top (Right). 40 trials are shown for one example neuron in each layer. (B) Left:
Average firing rates from the simulation (Top) and experimental data (Bottom). Right: $d'$ for detecting
an input to ORNs at each layer, computed from the same data as Left. (C) Same plots as B, with
a model with differentiator PNs (Top), and with integrator LHNs (Bottom). (D) Spectral power
amplification, $P_{LHN}(\omega)/P_{ORN}(\omega)$, normalized by total power. $P(\omega)$ is a power spectral density of
mean firing rate. (E) Information transfer from ORNs to PNs (solid) and LHNs (dotted). Black dotted
lines represent 100% information transfer. Grey regions and stars represent frequency bands with
significant differences between PNs and LHNs (*: $P<0.01$, Student $t$-test). Data are mean±SEM.

experimental measurements for i) the mean spontaneous firing rates in all layers and higher cell-to-cell
variability in PN firing rates (Jeanne and Wilson, 2015), ii) mean peak firing rates, and iii) rate of
decrease in mean LHN firing rate.

In the simulation of the optogenetic experiments in Jeanne and Wilson (2015), our network model
reproduced the key features of experimental results. When ORNs were given a common current input
that simulates optogenetic stimulation in experiments (Fig. 2(A)), PNs showed a slower amplified
response to transient inputs from ORNs, and LHN firing was more temporally refined, with the peak
of their firing rate preceding that of presynaptic PNs, just as in experimental data (Fig. 2(B) Left).
This rapid response of LHNs caused detection accuracy ($d'$) (see Jeanne and Wilson (2015) and
Appendix A2) for the ORN input to grow much faster to a larger maximum in LHNs than in PNs
(Fig. 2(B) Right).

We also constructed and tuned homogeneous network models, in which PNs and LHNs are of the
same type for comparison. In contrast, they showed suboptimal behaviors, such as the delayed firing
of LHNs; therefore, $d'$ of LHN rose more slowly and reached a lower maximum than that of PNs
(Fig. 2(C)).

How do the different intrinsic properties of neurons contribute to the speed and high fidelity of LHN
output? Since PNs and LHNs have opposite traits of differentiators and integrators, respectively,
their effects can compensate for each other in the combined feedforward transformation of the ORN
output. To analyze how the PN and LHN layers transform ORN inputs together, we computed how
they amplify the power spectrum of ORN firing within a physiological frequency band ($\leq 100$ Hz)
with data from another longer simulation with the continuous current stimulus to ORNs. The results
(Fig. 2(D)) showed that homogeneous networks with differentiator and integrator PNs/LHNs prefer-
entially amplified higher or lower frequency components, respectively, whereas the heterogeneous
network showed little distortion across the entire frequency range, demonstrating that PNs and LHNs
compensated distortions introduced by each other.

We found that this mechanism also facilitated information transfer. We estimated (the lower bound of) mutual information (MI) between the input to ORNs and spike outputs of each layer (Borst and Theunissen, 1999) (see also Appendix A2). In this way, we compared how much information in ORN firings pertaining to the input is transmitted to the output firing of PNs and LHNs. Specifically, we measured the information transfer from ORNs to PNs or to LHNs by computing a ratio of MIs, $I(\text{ORN input}; \text{PN or LHN output})/I(\text{ORN input}; \text{ORN output})$, respectively, where $I(X;Y)$ denotes MI between $X$ and $Y$. We found that information transfer to PNs closely matched that to LHNs in the heterogeneous network, whereas significant information loss was observed in homogeneous networks (Fig. 2(E)). In particular, the all-integrator PN/LHN case showed information loss specifically in the high-frequency band (Fig. 2(E), Right), indicating that large signal distortion in this regime (Fig. 2(D)) impaired information transfer. Interestingly, if we keep laminar heterogeneity but reversed the neuronal properties between PN and LHN, the network also shows stable power amplification and good information transfer (Appendix A3). These showed that laminar-specific intrinsic and functional properties of PNs and LHNs enabled nearly optimal information transfer between those neurons.

### 4.3 Lamina-specific neuronal properties promote robust and stable signal propagation in deep FFNs

We then investigated whether this mechanism can also enhance signal transmission in larger networks. For this purpose, we extended the AL network to a deep heterogeneous FFN model by adding more alternating layers of integrator or differentiator neurons. The deep FFN models had nine layers of 1,000 differentiator or integrator neurons in the AL network model, except for the input layer comprising differentiators. Again, each neuron was randomly connected to nine presynaptic neurons on average. Synaptic conductances and other parameters were the same as the AL network. The network parameters can be found in Appendix Table A1, A2.

We simulated how a packet of spikes, injected into the input layer, propagates through subsequent layers (Diesmann et al., 1999; van Rossum et al., 2002; Reyes, 2003; Kumar et al., 2008a, 2010; Kremkow et al., 2010; Moldakarimov et al., 2015; Joglekar et al., 2018). Spike generators randomly sampled in total $\alpha$ spike times from a normal distribution with variance $\sigma^2$ and forced the input layer neurons to fire at the spike times, in addition to noisy spontaneous firing.

We found that the spike signals stably propagated in this network, whereas homogeneous networks, with only differentiators or integrators, showed opposing results (Fig. 3(A,B)): In the all-differentiator network, the evoked spike signal became increasingly synchronized and propagated as layer-wide synchronized spikes. In contrast, in the all-integrator network, the evoked spike signal became broader and less synchronized, until it eventually dissipated into spontaneously firing spikes (Fig. 3(A) Right). Stable propagation in the heterogeneous network was decidedly robust over a wide range of input signals with diverse temporal width ($\sigma$) and the total number of spikes ($\alpha$) (Fig. 3(C) Top). Conversely, the all-differentiator network exhibited a clear preference for sharply synchronized spikes, while signals gradually dissipated into spontaneous activity in the all-integrator network (Fig. 3(C) Middle-Bottom). Therefore, when tested with input signals with diverse ($\sigma$, $\alpha$), the heterogeneous network showed the best performance in signal propagation (Appendix Fig. A2), and this result did not significantly change with additional feedforward inhibition in the deep FFN (Appendix Fig. A2).

Furthermore, we examined the case in which the input layer spiked with the dynamically fluctuating firing rate, due to dynamical, stochastic current injection (the same OU process as the AL network, Equation 4, except that $\mu_{input}$=25 $\mu$A/cm$^2$ and $\sigma_{input}$=12.5 $\mu$A/cm$^2$. See Appendix Table A3 for the other parameters). This continuous signal propagated in the heterogeneous network with many conserved features, whereas significant signal distortion and loss were again observed in the homogeneous networks (Fig. 3(D)). Note that propagation of dynamical input features indicates superior information transfer in a heterogeneous network, compared to homogeneous ones.

Again, the robust and stable signal propagation was possible by the distortion-compensating input/output transformations by neighboring layers with distinct neuron types. To demonstrate this, we analyzed trajectories of propagating signals in the ($\sigma$, $\alpha$) plane (Diesmann et al., 1999; Kumar et al., 2008b) (Fig. 3(E)), a simple version of the signal space that we previously discussed (Fig. 1). In the heterogeneous network, each layer transformed an incoming signal into a different, sometimes nearly opposite or complementary direction in the ($\sigma$, $\alpha$) plane than those transformed by its pre- and postsynaptic layer, which prevents the formation of a uniform flow. Therefore, a propagating

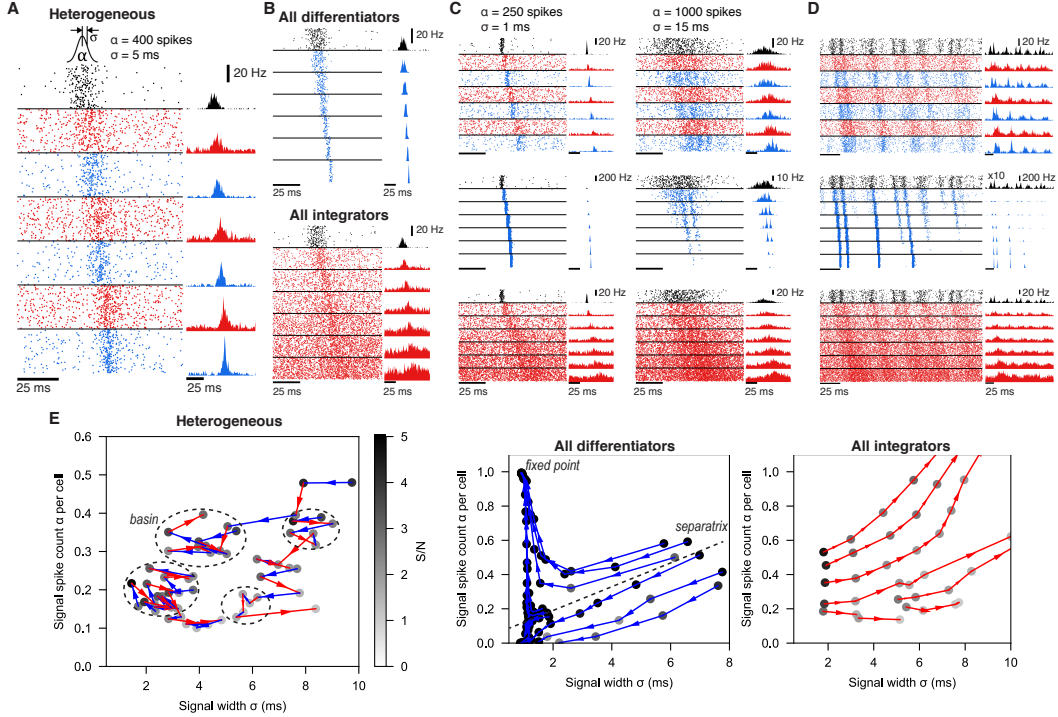

Figure 3: **Lamina-specific neuronal properties robustly stabilize spike signal propagation in deep FFNs.** (A) Propagation in a heterogeneous network. Inset on top is the Gaussian distribution of spikes, evoked in the input layers (black). Dots (Left) and histograms (Right) are spikes and their firing rates, respectively. In all figures, blue and red represent differentiators and integrators, respectively (B) Firing in homogeneous networks with only differentiators (Top) and integrators (Bottom). (C) Propagation of signals with different spike count ($\alpha$) and width ($\sigma$). (D) Network firing with continuous noise current in the input layer. In the middle row (blue; all differentiator), the input layer firing rate is multiplied by 10 for clarity. (E) Analysis of signal transformations underlying stable propagation in the ($\sigma$, $\alpha$) space. Each trajectory is formed by connecting ($\sigma$, $\alpha$) of a propagating signal (dots) between adjacent layers, starting from the second layer output. Shade of each dot is the signal-to-noise ratio (S/N) and only points with S/N>1 are shown. Dotted circles mark "basins" (Fig. 1(B)) where any propagating signal stays for $\geq$5 layers. A dotted line in the Middle panel is an approximated separatrix between trajectories toward a fixed point and dissipation (Fig. 1(A)). All models have 9 layers and the first 7 layers are shown in a-d for clarity.

signal cannot run away and is confined to a small region (basin), corresponding to stable propagation (Fig. 3(E) Left). However, in homogeneous networks, all layers perform similar transformations and drive propagating signals rapidly toward a fixed point of sharp synchronization or dissipation (Fig. 3(E) Middle, Right). Notably, in most of the ($\sigma$, $\alpha$) plane, transformations in those two networks are in nearly opposite directions: In the all-differentiator network, $\sigma$ and $\alpha$ both tend to decrease (Fig. 4e Middle), because sharply correlated spikes are the preferred input of the neurons, while $\sigma$ and $\alpha$ increase in the all-integrator network (Fig. 3(E) Right). In the heterogeneous network, those two different transformations are performed by neighboring layers to minimize overall signal distortion and boost information transfer. In summary, the distortion compensation mechanism by distinct neuron types can protect a propagating signal from undergoing a loss or distortion regime in the signal space, while supporting the robust and stable transmission.

## 4.4 Signal amplitude-dependent propagation latency in the heterogeneous FFN

While many studies have investigated the stable propagation of spike signals, the question of how rapidly the signals propagate through layers has received relatively less attention. However, millisecond-level signal transmission latencies in neural circuits are known to play crucial roles (Van-Rullen et al., 2005; Chechik et al., 2006), e.g. in sound localization by birds and mammals using

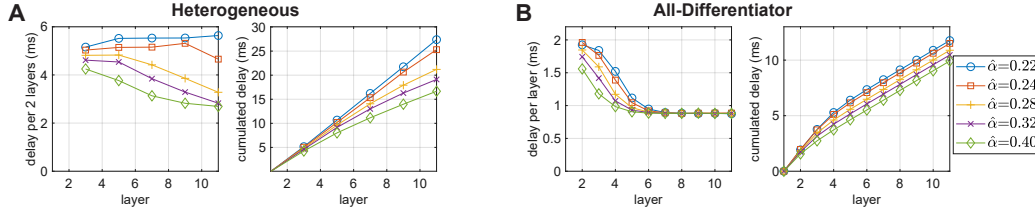

Figure 4: Propagation latency of spike signals in the (A) heterogeneous and (B) all-differentiator FFN. We used the input signals with $\sigma = 1$ ms for different firing rate, which therefore had the different numbers of spikes per cell in the input layer, $\hat{\alpha}$. The delays were computed as the differences of the median spike time $t_c$ of the signals between layers (see Appendix A2).

interaural time differences (Grothe et al., 2010). The propagation latency depends on the properties of synaptic connections, such as dendrite length, myelination, etc. In deep FFNs, the features of a propagating signal can also determine its propagation latency, since a transmission latency from one layer to another depends on how rapidly presynaptic spikes trigger spiking in the postsynaptic neurons, a property governed by their intrinsic mechanisms.

To examine this, we simulated our deep FFNs, now with more neurons (10,000) in each layer (and reduced the connection density correspondingly to keep each neuron randomly connected to 9 presynaptic neurons on average) for better statistical testing (Fig. 4). The results show that in the heterogeneous FFN, the propagation latency in deep layers varied with the input signal amplitude $\alpha$ (Fig. 4(A)). In the case of all-differentiator FFN, the propagation delay varied only for a few initial layers and quickly converged in deeper layers (Fig. 4(B)). In the case of the all-integrator FFN, the quick dissipation of spike signals (Fig. 3(B-D)) made the measurement of signal latency meaningless.

This result suggests that lamina-specific cell types in deep FFNs can naturally lead to neurons using both of the spike rate and time coding for an input signal (Panzeri et al., 2010; Ratté et al., 2013), especially in deeper layers. Therefore, deeper layers can develop more complex and expressive representations of network inputs. It will be of interest to further investigate the emergent coding properties in the deep spiking neural network, as a model of the neural systems, and their cellular basis.

## 5   Discussion

Diversity of cell types is one of the distinctive characteristics in neural systems, and its functional characterization is the subject of ongoing experimental investigations (e.g., Gouwens et al. (2019)). Integrating information about cell types and their intrinsic properties with network connectivity should be a pivotal research question to develop a holistic understanding of how spike signals propagate in neural circuits. However, the diversity of cellular properties is one of the most neglected elements in theoretical neural network studies.

We focused on functionally distinct cell types due to different voltage-dependencies of $K^+$ channels, which can arise from diverse expression patterns of low-threshold $K^+$ channels (Higgs and Spain, 2011; Hu et al., 2007; Svirskis et al., 2002). However, other neuronal mechanisms that affect the integrative cellular property can play similar roles, such as morphology (Mainen and Sejnowski, 1996), inactivation of $Na^+$ channels (Arsiero et al., 2007; Mensi et al., 2016), h-channels (Kalmbach et al., 2018), the high-conductance state (Destexhe et al., 2003; Prescott et al., 2008), etc. Furthermore, synaptic and circuit mechanisms known to operate as integrators or differentiators can be organized by a similar principle. For example, short-term synaptic depression and facilitation can act as high- or low-pass filters, respectively (Izhikevich et al., 2003), and inhibition can limit the integration time window for incoming inputs and promote temporal fidelity of neuronal responses (Pouille and Scanziani, 2001). Our hypothesis predicts that integrator neurons, such as PNs, tend to have synapses with short-term depression (Kazama and Wilson, 2009) whereas differentiators, such as LHNs, have facilitating synapses.

Jeanne and Wilson compared spike signal transfer from thalamocortical to cortical layer IV neurons to that between PNs and LHNs (Jeanne and Wilson, 2015). Likewise, we further propose that these

theoretical mechanisms can be applied to the thalamocortical loop and cortico-cortical feedforward projections, where spike signals propagate through multiple types of principal neurons that are different in size, morphology, ion channel expressions, etc. for each layer. Stable signal propagation in an FFN has been extensively studied in this context (Diesmann et al., 1999; Joglekar et al., 2018; Kremkow et al., 2010; Kumar et al., 2008a, 2010; Moldakarimov et al., 2015; Reyes, 2003; van Rossum et al., 2002; Vogels and Abbott, 2005). However, proposed models so far were often successful only with a limited range of input signals given fixed model parameters, although some precisely tuned models can handle a diverse range of inputs (Kumar et al., 2008a, 2010; van Rossum et al., 2002; Vogels and Abbott, 2005). In this study, we proposed a novel approach to this problem, based on an information-theoretic perspective, pointing out that an assumption of a single cell type in a network can result in accumulated signal distortion. Introducing multiple cell types with lamina-specific neuronal properties can circumvent this problem, exemplified by stable propagation of a dynamical spike signal in our model. Given the prevalence of diverse cell types in many neural systems, our work presents a clear case that lamina-specific cell types are critical to understanding network functions.

If pre- and postsynaptic neurons have differences in the intrinsic integrative properties, it inevitably causes a mismatch in their input/output transformations, and we suggest that this phenomenon can be a signature of optimal information transfer. A previous study suggested a different interpretation that the postsynaptic layer filters out a substantial fraction of information encoded by an input layer if such a mismatch is present (Blumhagen et al., 2011). PNs and LHNs in the *Drosophila* AL network also have mismatching input/output transformations due to differences in their intrinsic properties. However, our model showed that the information transfer from PNs to LHNs was nearly optimal, instead of LHNs filtering down a significant fraction of the information carried in the PN firing. Likewise, distinct stimulus encoding schemes between pre-/postsynaptic neurons, ubiquitously found in many neural systems, can be a mechanism for boosting information transfer across the entire network. Although not identical to our case, there is an analog in artificial deep neural networks, the *Residual Network* (He et al., 2016), which significantly outperforms conventional deep networks consisting of homogeneous convolutional layers. The residual network uses two types of transmission functions (a parameterized mapping $f(x)$ and a shortcut connection $x + f(x)$), which allows gradients to backpropagate across deep layers without vanishing or exploding (He et al., 2016).

Performing compensatory signal transformation is a widely used strategy in information-theoretic algorithms for optimizing information transfer with limited bandwidth, such as water-filling (Gallager, 1968). In this study, we have demonstrated how this scheme operates in FFNs when lamina-specific neuron types have different intrinsic properties. Notably, functionally different cell types within a layer can also be explained by the maximization of information transmission (Kastner et al., 2015). Therefore, we suggest that the commonly observed diversity of cell types in neural circuits is essential to achieve optimal information transmission.

## Broader Impact

Many efforts have been paid to understand the critical components of highly cognitive systems like the human brain. Studies have argued for simulations of large brain-scale neural networks as an indispensable tool (De Garis et al., 2010). Still, they almost always fail to consider cellular diversity in the brain, whereas more and more experimental data are revealing its importance. Our computational study suggests that heterogeneity in neuronal properties is critical in information transfer within a neural circuit and it should not be ignored, especially when the neural pathway has many feedforward layers.

For deep-learning research, our work also provides a new insight for neural architecture search (NAS) (Elsken et al., 2019). The search space of existing NAS methods are mainly (1) the combination of heterogeneous layers to form an entire network; (2) the combination of heterogeneous activation functions to form a cell. However, our work suggests a novel, computationally efficient strategy, that is searching for a block structure consisted of several layers (In our case, the block is composed of a integrator layer followed by a differentiator layer). On the one hand, the block should boost stable propagation of input signals into deep layers. Hence, divergence of inputs will remain detectable in the output layer at the initial phase of learning, which is suggested to accelerates the training of very deep networks (Samuel S Schoenholz and Sohl-Dickstein, 2017; Srivastava et al., 2015). On

the other hand, there is also extra freedom of searching the block structure that does not suffer from vanishing/exploding backpropagation gradients (like a residual block (He et al., 2016))

Our deep FFN models are proof-of-concept and lack many other neural circuit mechanisms that can affect signal propagation in spiking neural networks, as we discuss in Section 2, although we did find that an additional component, feedforward inhibition, did not significantly change the results (Appendix Fig. A2,A3). Our study suggests that the cooperation between different types of neurons is vital for promoting signal processing in large-scale networks. It also suggests investigating the roles of heterogeneous neuronal properties in other problems such as sensory coding, short-term memory, and others, in the future studies.

## Acknowledgements

The authors declare no conflict of interest. We thank James Jeanne for helpful discussions and for sharing experimental data. We also thank Steven Prescott, Mario Negrello, Jihwan Myung, and Steven Aird for reading an earlier version and providing useful feedback. This work was supported by funding from the Okinawa Institute of Science and Technology Graduate University. SH was also supported by Japan Society for the Promotion of Science, KAKENHI Grant Number 15K06725.

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
