[Supplementary Material]

## Appendix A1  Dynamical spiking threshold of the differentiator neurons

The dynamic threshold is a neural property where firing not only depends on the membrane potential but also on its temporal change, which is crucial for neural sensitivity to input fluctuations (Prescott et al., 2008; Azouz and Gray, 2003; Hong et al., 2008). An experimental study showed that LHNs have dynamic thresholds (Jeanne and Wilson, 2015), and differentiator neurons are also known to have the property. Therefore, we used differentiator neurons for modeling LHNs, and integrators with for PNs in the AL network. To demonstrate their difference in the spiking threshold, we estimated a minimal rate of membrane potential change, $[dV/dt]_{\text{min}}$, that preceded spikes, but not subthreshold fluctuations from our simulation data, which corresponds to the minimal inward current required for spiking (Fig. A1(B)). Then, the threshold voltage, $V_\theta$, at $dV/dt \approx [dV/dt]_{\text{min}}$, was significantly more distributed in differentiators (Integrator: STD[$V_\theta$]=2.76±0.08 mV, Differentiator: 3.33±0.11 mV; $P$=2.58×10$^{-5}$, $F$-test; Fig. A1(C)), suggesting that differentiators can generate enough inward current to generate a spike across a broader range of membrane voltages than integrators, an indication of a more dynamical spiking threshold. Therefore, we used differentiator neurons, with the low-threshold K$^+$ channel, for modeling LHNs, and integrators with the high-threshold channel for PNs in the AL network.

## Appendix A2  Data analysis

In the AL network model case, $d'$, a measure for signal detection, was computed in the same way as in (Jeanne and Wilson, 2015):

$$d' = \frac{\mu_{stim} - \mu_0}{\sqrt{(\sigma_{stim}^2 + \sigma_0^2)/2}} \tag{1}$$

where $(\mu_{stim}, \sigma_{stim})$ and $(\mu_0, \sigma_0)$ are the (mean, STD) of spike count at a given layer, computed with 80 ms-long overlapping temporal windows in the stimulated and non-stimulated condition, respectively. For each layer, we computed $d'$ of all the cells and plotted their median in Fig. 2(B,C).

Power spectra for Fig. 2(D) were evaluated by applying the MATLAB function `pmtm` with a 20-ms time window on spike trains formed with 1-ms time bins. Mutual information in Fig. 3E were computed by a Gaussian channel approximation (Borst and Theunissen, 1999): We first reduced the dimensionality of a population spike trains at each layer, by using principal component analysis (PCA). Since the first PCA component was always dominating, we projected the population spike trains to this component to form a one-dimensional "population response" time series. With the Fourier transformation of the stimulus and population response, $S(\omega)$ and $R(\omega)$, we estimated a kernel $K(\omega) = < R^*(\omega)S(\omega) > / < R^*(\omega)R(\omega) >$, and computed a reconstructed stimulus and noise via $Sr(\omega) = R(\omega)K(\omega)$ and $N(\omega) = S(\omega) - Sr(\omega)$. The mutual information per each frequency bin was then computed by

$$I(S(\omega); R(\omega)) = \log_2(1 + SNR(\omega)), \quad SNR(\omega) = \|S_r(\omega)\|^2/\|N(\omega)\|^2. \tag{2}$$

With this, we computed the information transfer (Fig. 3E) by

$$T_X(\omega) = I(S(\omega)_{\text{ORN}}; R(\omega)_X)/I(S(\omega)_{\text{ORN}}; R(\omega)_{\text{ORN}}),$$

where $X$ is PN or LHN.

In the deep FFN, we computed $(\sigma, \alpha)$ for spikes from each layer using a custom algorithm that estimates $(\sigma, \alpha)$ in the presence of additional spontaneous firing. We first computed the baseline spontaneous firing rate $\nu_0$ at each layer by averaging the firing rate obtained from the same model with no input. The firing rate curve was computed by histogramming spike times in this layer with a 0.1-ms time bin and by smoothing it with a 3-step moving average. Then, we evaluated a least-square fit of $\nu(t)$ to $\nu_{fit}(t) = \nu_0 + \nu_1 \exp(-(t - t_c)^2/2\sigma^2)$. $\alpha$ was estimated by counting the spikes in the $[t_c - 3\sigma, t_c + 3\sigma]$ window. From the goodness of fit, $R^2 = 1- < (\nu(t) - \nu_{fit}(t))^2 > /\text{Var}[\nu(t)]$, we evaluated the signal-to-noise ratio, $S/N = R/(1 - R^2)^{1/2}$ (Fig. 3(E)).

All analysis was performed by custom codes written in MATLAB 2016b (MathWorks, MA) and Python. All the models are publicly available at `https://github.com/FrostHan/HetFFN-`. The datasets generated during and/or analyzed during the current study and analysis codes are available from the corresponding author upon reasonable request.

## Appendix A3  Reversed heterogeneous FFNs and intermediate $\beta_w$

In the AL network, we modeled PNs as integrators and LHNs as differentiators according to the experimental findings (Jeanne and Wilson, 2015). As alternative cases, we simulated the AL network of homogeneous PNs and LHNs with intermediate $\beta_w$=-10 mV, and the same network with the properties of PNs and LHNs reversed (i.e. PNs and LHNs are differentiators and integrators, respectively). The results of homogeneous AL network with intermediate $\beta_w$ are shown in Fig. A4((A-C)) Left and D. Compared with the original heterogeneous AL model (Fig. 2), the homogeneous AL network performs worse in terms of accuracy and information transfer (Fig. A4((B-C) Left), and shows less stable power amplifcation (Fig. A4((D)). The reversed AL network shows the stable power amplification and good information transfer (Fig. A4((C)) Right, and (D)). However, $d'$, accuracy of the ORN input detection, is suboptimal (Fig. A4 (B) Right; dots are lower than solid lines), since ORNs fire sparsely with strong differentiator characteristics (Nagel and Wilson, 2011) and, therefore, integrators can be better suited for their postsynaptic cells.

We further extended these two kind of network models to deep FFNs (Fig. A5). In the homogeneous FFNs with intermediate $\beta_w$ = -7 mV, -10 mV and -12 mV, a relatively weak and asynchronous spike signal dissipated in deep layers (Fig. A5 A-C), while a relatively strong and synchronous one tended to diverge (Fig. A5 E-G). In contrast, the reversed heterogeneous network showed robust and stable signal transmission (Fig. A5 D, H), similar to our original heterogeneous FFN model. Therefore, stable signal propagation can be achieved only by proper demodulation of signal distortions between adjacent layers, regardless of the order of integrator / differentiator. In contrast, if an FFN contains only homogeneous layers, the signal propagates with accumulated distortion (amplification/dicrease in $\alpha/\sigma$) into deep layers.

## Appendix A4  Sensitivity to $\beta_w$ in deep heterogeneous FFNs

$\beta_w$ for integrator (5 mV) and differentiator ($-19$ mV) neurons in the deep heterogeneous FFNs, together with other hyper-parameters, were from the AL network model. Will changing $\beta_w$ impair stability of signal propagation in the heterogeneous network? We show a sweep of $\beta_w$ for integrator ($3, 5, 7$ mV) and differentiator ($-20, -19, -18$ mV) neurons in the deep heterogeneous FFN (Fig. A6).

Notably, varying $\beta_w$ for integrator neurons did not significantly affect the propagation property, while a different value of $\beta_w$ for differentiators changed the property. This is because the mean spike threshold increased significantly when $\beta_w$ of differentiator became smaller (Ratté et al., 2013). Therefore, the range of input ($\alpha, \sigma$) that can stably propagate reduced when $\beta_w = -20$ mV for differentiators (Fig. A6 A, D, G), and enlarged when $\beta_w = -18$ mV for differentiators (Fig. A6 C, F, I). However, we could compensate for the change in the spike threshold of differentiators, which was introduced by varying $\beta_w$, by increasing or decreasing the synaptic conductance from integrators to differentiators. Fig. A6 (J-K) shows that a similar propagation property to the original heterogeneous FFN model can be achieved by doing so.

As we discussed in Section 4.3, the robust and stable signal propagation is due to the distortion-compensating input/output transformations by neighboring layers with distinct neuron types. As long as this mechanism is not profoundly compromised, spike signals should be able to transmit stably and robustly.

Figure A1: **Intrinsic properties of conductance-based model neurons control dynamicity of spiking thresholds.** (A) Membrane potential response (color) to constant or fluctuating current injection (black). (B) Example membrane potential $V$ vs. $dV/dt$ in two neurons, based on simulation data in Fig. 2(A,B). Data from one trial are shown (gray). Dotted lines represent $[dV/dt]_{min}$, the minimal $dV/dt$ for spiking, and colored dots are threshold-crossing points. (C) Spread of membrane potentials at crossing points, $V_\theta$, from the average. Vertical bars span from 10% to 90% quantiles, and notches are at medians. Data are the same as B, and only 50 samples (dots) are shown for clarity.

| Parameter | Value |
|---|---|
| $E_{Na}$ | 50 mV |
| $E_K$ | -100 mV |
| $E_L$ | -70 mV |
| $g_{Na}$ | 20 mS/cm$^2$ |
| $g_K$ | 20 mS/cm$^2$ |
| $g_L$ | 2 mS/cm$^2$ |
| $\phi_w$ | 0.15 |
| $C$ | 2 mF/cm$^2$ |
| $\beta_w$ | -1.2 mV |
| $\gamma_m$ | 18 mV |
| $\gamma_w$ | 10 mV |
| $E_{syn}$, excitatory | 0 mV |
| $E_{syn}$, inhibitory | -90 mV |

Table A1: **Parameters of the single-neuron model.**

Figure A2: **Propagation of spike signals with diverse width ($\sigma$) and number of spikes ($\alpha$) in the heterogeneous (A), all-differentiator (B), and all-integrator network (C).** Each network has nine layers of 5,000 neurons (see Table A3 for parameters). Color in the middle column represents propagation depth, computed by numbers of layers (except the input layer) into which spike signals propagate. Propagation is considered stopped if the estimated $\alpha$ is lower than 0.05n or larger than 3n for a layer and its corresponding postsynaptic layer, where n=5,000 is the group size. Side insets are example raster plots for parameters marked by dotted squares in the middle, showing spikes from 10% of neurons at each layer for clarity.

| Parameter | Heterogeneous | Differentiator PN | Integrator LHN |
|---|---|---|---|
| $\beta_w$, ORN | -23 mV | -23 mV | -23 mV |
| $\beta_w$, PN | 5 mV | -19 mV | 5 mV |
| $\beta_w$, LHN | -19 mV | -19 mV | 5 mV |
| $\sigma_V$, ORN | 38 $\mu$A/cm$^2$ | 38 $\mu$A/cm$^2$ | 38 $\mu$A/cm$^2$ |
| $\sigma_V$, PN | $38 + 15\eta$ $\mu$A/cm$^2$ | 15 $\mu$A/cm$^2$ | $38 + 15\eta$ $\mu$A/cm$^2$ |
| $\sigma_V$, LHN | 15 $\mu$A/cm$^2$ | 15 $\mu$A/cm$^2$ | $38 + 15\eta$ $\mu$A/cm$^2$ |
| $g_{syn}$, PN | 345 $\mu$S/cm$^2$ | 1170 $\mu$S/cm$^2$ | 345 $\mu$S/cm$^2$ |
| $g_{syn}$, LHN | 975 $\mu$S/cm$^2$ | 715 $\mu$S/cm$^2$ | 285 $\mu$S/cm$^2$ |

Table A2: **Parameters of the *Drosophila* AL network model.** $\eta$ is a random number sampled from a uniform distribution ranging from 0 to 1.

Figure A3: **The same figures as Fig. A2, but using FFN models with feedforward inhibition.** Again, each network has nine layers of 4,000 PN-like or LHN-like excitatory neurons, combined with 1,000 inhibitory neurons that receive excitatory inputs from a previous layer and inhibit excitatory neurons in the same layer. Inhibitory cells were also based on the Morris-Lecar model with $\beta_w$ = -15 mV while different $\beta_w$ did not cause any significant change in our conclusion. The reversal potential of inhibitory synapses was $E_{syn}$ = -90 mV and the conductance was 200 $\mu$S/cm$^2$. Also, we added a synaptic delay of 2 ms for all connections. Other parameters were the same as those for Fig. A2. In all panels, we plotted spikes from 10% of excitatory neurons at each layer for clarity.

| Parameter | Heterogeneous | Differentiator PN | Integrator LHN |
|---|---|---|---|
| $\beta_w$, Input | -23 mV | -23 mV | -23 mV |
| $\beta_w$, Even | -19 mV | -19 mV | 5 mV |
| $\beta_w$, Odd | 5 mV | -19 mV | 5 mV |
| $\sigma_V$, Input | 38 $\mu$A/cm$^2$ | 38 $\mu$A/cm$^2$ | 38 $\mu$A/cm$^2$ |
| $\sigma_V$, Even | 15 $\mu$A/cm$^2$ | 15 $\mu$A/cm$^2$ | 38 + 15$\eta$ $\mu$A/cm$^2$ |
| $\sigma_V$, Odd | 38 + 15$\eta$ $\mu$A/cm$^2$ | 15 $\mu$A/cm$^2$ | 38 + 15$\eta$ $\mu$A/cm$^2$ |
| $g_{syn}$, Even | 975 $\mu$S/cm$^2$ | 975 $\mu$S/cm$^2$ | 345 $\mu$S/cm$^2$ |
| $g_{syn}$, Odd | 345 $\mu$S/cm$^2$ | 975 $\mu$S/cm$^2$ | 345 $\mu$S/cm$^2$ |

Table A3: **Parameters of the deep FFN model**. Even and Odd represent the $2n$ and $(2n + 1)$-th layer where $n = 1, 2, \ldots, 5$, respectively. $\eta$ is a random number sampled from a uniform distribution ranging from 0 to 1.

Figure A4: (A-C) Firing rates (A), $d'$ (B), and information transfer (C) for the homogeneous AL network with $\beta_w$=-10 mV (left) and reversed heterogeneous model (right). (D) Power amplification of the original (black), reversed (green), and $\beta_w$=-10 mV network model. Blue: Differentiator, Red: Integrator, Magenta: $\beta_w$=-10 mV. Shade in C: $P$<0.01.

Figure A5: Raster plot of deep homogeneous FFNs with intermediate $\beta_w$, and the deep reversed-heterogeneous FFN. The input spike signal is featured with ($\alpha$=400 spikes, $\sigma$=5 ms) for the first row (relatively weak and asynchronous input signal) and ($\alpha$=900 spikes, $\sigma$=2 ms) for the second row (relatively strong and synchronous input signal).

Figure A6: (A-I) Propagation of spike signals with diverse width ($\sigma$) and number of spikes ($\alpha$) in the heterogeneous networks with changed values of $\beta_w$ for differentiator (the first voltage) and integrator neurons (the second voltage) while keeping all other parameters unchanged. Plotted in the same way as Fig. A2. (J) Same as D but with larger conductance (1000 $\mu$S/cm$^2$) from integrators to differentiators, so as to reimburse the higher spike threshold introduced by smaller $\beta_w$. (K) Same as F but with smaller conductance (820 $\mu$S/cm$^2$) from integrators to differentiators.