[Reviews · NeurIPS 2020]

Review 1

Summary and Contributions: This paper redid the classic Ad Aertsen paper in which they asked whether synchronous volleys of spikes could propagate through multiple layers of a network. Unlike the original work, the goal here was to preserve the temporal width and firing rate of the synchronous volleys. By alternating neuron types in adjacent layers, they largely succeeded.

Strengths: The main strength is that they got it to work.

Weaknesses: The weakness is that it's not clear why this is important. If the brain wants to send information, it will probably use axons; if multiple layers are involved, it's doing some sort of computation. Update post-author-response: In their reply, the authors pointed out that their method would be used for non-connected areas (since the brain presumbably can't route new axons). I'm skeptical: it would have to route information through connected areas, using axons that are already used for something else. That seems insanely hard. Also, I'm worried about robustness. By using two neuron types, when the width (sigma) and amplitude (a) of a pulse go through two layers, they are transformed as [a_(l+2), sigma_(l+1)] = f(g(a_l, sigma_l)). where both f and g are vector functions with two outputs. In their simulations, f(g(a, sigma)) \approx [a, sigma]. More accurately, there seemed to be several fixed points in the [a, sigma] plane. This is non-generic behavior, and one wonders if there was fine tuning involved. Update post-author-response: the authors said no fine tuning, and pointed to Fig. 3E. To be convinced, I would need to see a parameter sweep.

Correctness: I think so.

Clarity: Yes.

Relation to Prior Work: Yes.

Reproducibility: Yes

Additional Feedback: None.


Review 2

Summary and Contributions: Feedforward networks (FFNs) with heterogeneous laminar-specific neuronal properties are shown to stabilize and enhance information transmission compared to homogeneous FFNs. Simulations results captured key experimental properties of the multilayered circuitry of the Drosophila olfactory system. The interplay between integrator and differentiator neuron types across network layers compensates distortions introduced by each other and provides robust spiking information transfer. Having read the author's reply and the other comments, I stand by my original assessment.

Strengths: The work provides simulation results that extensively describes how a particular heterogeneous multilayered circuitry could improve accuracy and speed of information transfer in comparison with the equivalent homogeneous circuitry. It raises the point that heterogeneity in properties of neuron types is a critical factor for robust information transfer, in particular when considering multilayered feedforward networks.

Weaknesses: The main conclusions of this work are particularly well suited to the experimental findings of the specific multilayered circuitry of the Drosophila olfactory system. However, it is not clear how general these conclusions are and if they could be extrapolated to other brain areas and other widely-studied neural architectures. For example, inhibitory neurons play an essential role in cortical processing but this work partially addresses their influence including only some results with some sort of unspecified feedforward inhibition in Appendix Fig. A

Correctness: Methodology is well-defined, proven, and thoroughly documented. Claims are correct based on the proposed methodology.

Clarity: The paper is well written.

Relation to Prior Work: It is clearly discussed how this work differs from previous modelling studies of FFNs. However, it would be interesting to note that although previous works did not explicitly model heterogeneity across layers, they did introduce heterogeneity into the network choosing parameters of neurons (e.g., Vth or C) from probabilistic distributions (see for example the work by Kumar and colleagues, 2008, the Journal of Neuroscience).

Reproducibility: Yes

Additional Feedback:


Review 3

Summary and Contributions: This paper uses the Drosophila’s olfactory system (ORN to PN to LHN) as an example to demonstrate how the heterogeneity in the intrinsic parameters of single neurons could benefit for information transmission in a feedforward network, which is the main conceptual contribution of this work. Then they extend the idea of heterogeneous neurons to multiple layers of feedforward networks.

Strengths: The authors did numerical simulations of the network model to support their claims. In particular, the network parameters are consistent with real neural circuit, in that they were fitted from experimental data. The simulation results are solid and clearly demonstrate with a repetition of integrator-differentiator networks, the firing rate profile of input spikes could be reliably transmitted through multiple layers (Fig. 3).

Weaknesses: Although I believe the intrinsic difference of neurons could benefit for information transmission, I have some conceptual questions. I think properly answer these questions in the Discussion or briefly mention some of them in author feedback could improve the impact of this work in general. 1. Whether a network is an integrator of a differentiator is highly determined by the value of \beta. Is it possible with an intermediate value of \beta, the network’s output is proportional to the input, i.e., the network simply relay the input but neither differentiating or integrating. In this case, we probably only need one layer to transmit the input without the cascade of an integrator and a differentiator. Update after rebuttal: I think in principle there might be a particular \beta to achieve same transmission with integrator-differentiator circuit, but some fine-tuned mechanisms are needed. And hence the heterogeneity would be a more robust mechanism to achieve reliable transmission. 2. If we reverse the order of differentiator and integrator, would the input be reliably transmitted as well? Is there some considerations for Drosophila’s olfactory system has a structure of an integrator followed by a differentiator? Update after rebuttal: I agree author’s statement about this and I hope the author could briefly discuss the order of integrator and differentiator in a revised manuscript.

Correctness: My own concern is about the conclusions based on Fig. 4 (correct me if I was wrong). How does Fig. 4A could lead to the conclusion that the proposed network model could adopt both spike rate and spike timing coding (line 228)? Update after rebuttal: I am still not clear about the justification of this. Please provide more elaborate claims in a revised manuscript.

Clarity: This paper is well written, and structure-wise.

Relation to Prior Work: The comparison with previous work was mentioned in Discussion. The authors mentioned earlier models need fine-tuned to achieve reliable transmission, while in current study the stable transmission was achieved by heterogeneity on intrinsic parameters of single neurons. One thing is not very clear is that have previous network models considered the heterogeneity on the intrinsic parameters of neurons? If not, I hope the author could clearly mention this point in a revised version to better emphasize the contribution of this work.

Reproducibility: Yes

Additional Feedback: 1. I am quite confused about the dynamics (Eq. 1), although I believe the authors used a correct dynamical equation in their simulation. The thing puzzled me a lot is that there is no interaction at all be 2nd row and and 1st row in Eq. 1. Moreover, the 2nd row says the variable z could be either m or w, and why the authors write a separate expression of w in the left most on the 2nd row? 2. Fig. 2C and Lines 144-146: I am a little concern about whether the comparison between the heterogeneous network and homogeneous network is fair. In computation, I am wondering whether the \beta could be adjusted into an intermediate value and then the cascade of two homogeneous networks is the same as a homogenous network? 3. I will be very glad to see if authors could provide a figure demonstrating how the network varies from an integrator to a differentiator when the value of \beta changes. 4. I am also curious to know how sensitive the reliable transmission depends on the values of \beta in two consecutive layers.

[Author Response · NeurIPS 2020]

We thank all reviewers for carefully reading our manuscript and their valuable comments. Our response is as follows.

**Response to Reviewer #1**

>> *...it's not clear why this is important. If the brain wants to send information, it will probably use axons.*

We dispute "it will probably use axons" because adult brains cannot grow long-distance axons, which could only happen
as an evolutionary process. If, as a consequence of learning, a brain wants to transfer information reliably between two
areas that previously were not functionally connected, it can use the heterogeneous FFNs in this study. We will add this
argument to the Discussion.

>> *seemed to be several fixed points in the $(\alpha, \sigma)$ plane . . . non-generic behavior . . . if there was fine tuning involved*
The mechanism requires no fine-tuning. The signal transformations by neighboring layers (Fig. 3E left) need not be in
exactly opposite directions but can only be sufficiently different to prevent continuous flows (Fig. 3E middle, right).

**Response to Reviewer #2**

>> *However, it is not clear how general . . . inhibitory neurons play an essential role . . .*
We agree that inhibitory neurons are important. However, in more realistic contexts, the mechanism is mostly about
how the initial part of the network input propagates before other mechanisms, such as recurrent inhibition, activate (see
also Ref. [1]). An exception can be fast feedforward inhibition, so we tested different levels of feedforward inhibition,
and found that the main results do not change as long as it is not too strong (e.g. Appendix Fig. A3).

>> *previous works . . . introduce heterogeneity into the network choosing parameters of neurons . . .*
Thanks for the suggestion. We will complement the discussion with this track of related works.

**Response to Reviewer #3**

>> *1. . . . Is it possible with an intermediate value of $\beta$ . . . the network simply relay the input . . .*
No. No matter what $\beta_w$ is used, it fixes the signal transformation property of neurons. Repeating the transform through
multiple layers will accumulate signal distortion and lead to information loss, as explained in Fig. 1 and related text.
We simulated the AL network model with neurons of $\beta_w$ =-10 mV (Fig. R1, left), and found that $d'$ for the ORN input
detection and information transfer are impaired compared to the original, heterogeneous model.

>> *2. If we reverse the order of differentiator and integrator, would the input be*
*reliably transmitted as well? . . .*
Yes. The reversed AL network also shows the stable power amplification and good
information transfer (Fig. R1A-C, right and D). In the deep network models, we
made similar observations. However, $d'$ for the ORN input detection is suboptimal
(Fig. R1B; dots are lower than solid lines), because ORNs fire sparsely with strong
differentiator characteristics (Ref. [17]). Integrators are better suited for ORN's
postsynaptic cells.

>> *. . . have previous network models considered the heterogeneity on the intrinsic*
*parameters of neurons?*
Some previous studies considered heterogeneous neuronal property, but not in a
laminar-specific manner. We will complement related discussion.

>> *I am quite confused about the dynamics (Eq. 1)...*
Thanks for carefully checking the equations. Eq. 1 had a typo, and the second
term of the right hand side should be $-g_K w(V - E_K)$, not $-g_K(V - E_K)$. The
correct equation was used in our source code. Note that $w_\infty$ is a function of $V$,
used for the equation for $w$. We will fix the equations in the revision.

>> *How does Fig. 4A lead to $\cdots$ the proposed model could adopt both spike rate*
*and spike timing coding . . . ?*
We meant that Fig. 4A corresponds to spike timing (latency), in addition to the
previous results (e.g. Fig. 2D) related to rate coding. We will clarify it in revision.

>> *Fig. 2C and Lines 144-146 . . . I am wondering whether $\beta_w$ could be adjusted*
*into an intermediate value . . .*
See our response to the question 1 and Fig. R1.

>> *. . . provide a figure demonstrating how the network varies . . . when $\beta_w$ changes*
We will happily add the figures requested by the reviewer in the next version.

>> *how sensitive the reliable transmission depends on the values of $\beta_w$ in two*
*consecutive layers.*
Please refer to our response to the second questions of Reviewer #1.

Figure R1: **A-C** Firing rates (A), $d'$ (B), and information transfer (C) for the homogeneous AL network with $\beta_w$=-10 mV (left) and reversed heterogeneous model (right). **D** Power amplification of the original (black), reversed (green), and $\beta_w = $ -10 mV network model. Blue: Differentiator, Red: Integrator, Magenta: $\beta_w$=-10 mV. Shade in C: *P*<0.01.

[Meta-Review · NeurIPS 2020]

The reviewers agree that this paper merits acceptance, though they do raise a number of important issues regarding the generality of these results and the sensitivity to parameter choices. Please update the paper to address these concerns, and if possible, include a parameter sweep/sensitivity plot in the supplementary material. Though the reviewers did not say it, I think Section 2: Related Work needs to be substantially expanded to situate and motivate this paper.